# Therapeutic Potential of Magnetic Nanoparticle-Based Human Adipose-Derived Stem Cells in a Mouse Model of Parkinson’s Disease

**DOI:** 10.3390/ijms22020654

**Published:** 2021-01-11

**Authors:** Ka Young Kim, Keun-A Chang

**Affiliations:** 1Department of Nursing, College of Nursing, Gachon University, Incheon 21936, Korea; kykim@gachon.ac.kr; 2Neuroscience Research Institute, Gachon University, Incheon 21565, Korea; 3Department of Pharmacology, College of Medicine, Gachon University, Incheon 21936, Korea; 4Department of Health Sciences and Technology, GAIHST, Gachon University, Incheon 21936, Korea

**Keywords:** Parkinson’s disease, human adipose-derived stem cells, magnetic nanoparticles, in vivo imaging

## Abstract

Parkinson’s disease (PD) is a progressive neurodegenerative disease characterized by the loss of dopaminergic neurons in the substantia nigra. Several treatments for PD have focused on the management of physical symptoms using dopaminergic agents. However, these treatments induce various adverse effects, including hallucinations and cognitive impairment, owing to non-targeted brain delivery, while alleviating motor symptoms. Furthermore, these therapies are not considered ultimate cures owing to limited brain self-repair and regeneration abilities. In the present study, we aimed to investigate the therapeutic potential of human adipose-derived stem cells (hASCs) using magnetic nanoparticles in a 6-hydroxydopamine (6-OHDA)-induced PD mouse model. We used the Maestro imaging system and magnetic resonance imaging (MRI) for in vivo tracking after transplantation of magnetic nanoparticle-loaded hASCs to the PD mouse model. The Maestro imaging system revealed strong hASCs signals in the brains of PD model mice. In particular, MRI revealed hASCs distribution in the substantia nigra of hASCs-injected PD mice. Behavioral evaluations, including apomorphine-induced rotation and rotarod performance, were significantly recovered in hASCs-injected 6-OHDA induced PD mice when compared with saline-treated counterparts. Herein, we investigated whether hASCs transplantation using magnetic nanoparticles recovered motor functions through targeted brain distribution in a 6-OHDA induced PD mice. These results indicate that magnetic nanoparticle-based hASCs transplantation could be a potential therapeutic strategy in PD.

## 1. Introduction

Parkinson’s disease (PD) is a progressive neurodegenerative disease characterized by the loss of dopaminergic neurons in the substantia nigra (SN) [1]. The clinical features of PD include motor symptoms such as resting tremor, bradykinesia, limb rigidity, and gait and balance problems, as well as non-motor symptoms such as sleep behavior disorder, apathy, mood changes, anxiety, constipation, and loss of olfaction. Several treatments for PD have focused on the management of physical symptoms using dopaminergic drugs [1,2,3]. However, these treatments cause various adverse effects including hallucinations and cognitive impairment owing to the non-targeted delivery to the brain despite alleviating motor symptoms of PD. Furthermore, these agents are not deemed ultimate cures for the disease itself owing to limited abilities of brain self-repair and regeneration [4].

Stem cell-based therapy as regenerative medicine has been highlighted in neurodegenerative diseases [5,6,7,8,9]. Stem cell-based regenerative therapy is known to improve neuroprotection and tissue repair ability in neurodegenerative diseases [5,10]. In particular, human adipose-derived stem cells (hASCs), a type of mesenchymal stem cells (MSCs), are obtained from abundant human adipose tissues and allow autologous transplantation, as well as possess the ability of neuron-like differentiation [11,12]. Moreover, the transplantation of hASCs improves behavioral performance and restores the loss of dopaminergic neurons in a PD animal model [10,13,14]. However, some clinical trials are reported to have little evidence that stem cell-based therapy is effective for PD patients [15,16]. In addition, the blood brain barrier (BBB) is still a major limitation for stem cell delivery to the brain [17].

Recently, the application of magnetic nanoparticles has been considered as potential delivery vehicles to improve the efficacy of stem cell-based therapy in neurodegenerative diseases [18,19]. Magnetic nanoparticles are powerful vehicles that can pass the blood-brain barrier and can be widely employed in the diagnosis and treatment of diseases, including as contrast agents for magnetic resonance imaging (MRI), drug delivery, or specific cell delivery and tracking [20,21]. Magnetic nanoparticles reportedly improve targeting of stem cells and efficacy of stem cell-based therapy [22,23,24]. Furthermore, magnetic nanoparticle-based hASCs tracking may be a major field for regenerative medicine, including PD.

Thus, the aim of this study was to investigate the therapeutic potential of hASCs using magnetic nanoparticles in a 6-hydroxydopamine (6-OHDA) induced PD mouse model.

## 2. Results

### 2.1. Experimental Procedures

To investigate the therapeutic potential of hASCs (ASC) using magnetic nanoparticles (MNP), we employed four mouse groups for behavior tests: saline-injected sham (Sham/Saline), MNP labeled hASCs-injected sham (Sham/ASC-MNP), saline-injected 6-OHDA-induced PD (6-OHDA/Saline), and MNP labeled hASCs-injected 6-OHDA-induced PD (6-OHDA/ASC-MNP) group. Furthermore, we performed with the following four groups (Sham/ASC-MNP, 6-OHDA/Saline, 6-OHDA/MNP and 6-OHDA/ASC-MNP groups) for MR images, and used the following four groups (Sham/Saline, Sham/ASC-MNP, 6-OHDA/MNP, and 6-OHDA/ASC-MNP groups) for Maestro imaging experiments. During the experimental schedule, baseline behaviors including body weight, apomorphine-induced rotation test, which measures the hypersensitivity of the lesioned striatum, and the rotarod test, used to assess motor coordination, were performed to evaluate the PD mouse model 1 week after 6-OHDA injection (2 days before hASCs injection) (Figure 1A). Then, to assess the motor recovery following hASCs transplantation, an apomorphine-induced rotation test and rotarod test were performed at 3 weeks (12 days after hASCs injection) and 6 weeks (33 days after hASCs injection) after 6-OHDA injection (Figure 1A). The 6-OHDA was stereotaxically injected into the SN (Anterior-Posterior (AP), −3.2 mm; Medial-Lateral (ML), −1.5 mm; Dorsal-Ventral (DV), −4.6 mm) (Figure 1B). To evaluate the distribution of hASCs, we labeled hASCs with magnetic nanoparticles (Magnoxide 797) which is magnetic silica nanoparticles (core-shell) containing Near-infrared (near-IR) fluorescent dye. MRI was performed on 1 day before and 2 and 6 days after hASCs transplantation (Figure 1A). In vivo cell tracking was performed using the Maestro imaging system on day 9 after hASCs transplantation.

One week after 6-OHDA injection, there was no body weight difference between sham and 6-OHDA PD group mice (Figure 1B). In baseline behavior tests, 6-OHDA-induced PD mice showed a significant increase in the number of rotational responses of the apomorphine-induced rotation test, and also had a reduced latency time falling from the rod rotated in the rotarod test compared to Sham mice (Figure 1C).

### 2.2. In Vivo Magnetic Resonance Imaging of hASCs in 6-OHDA Induced PD Mouse

MRI images were acquired to evaluate the distribution of magnetic nanoparticle (MNP)-based hASCs in Sham/ASC-MNP, 6-OHDA/Saline, 6-OHDA/MNP, 6-OHDA/ASC-MNP groups, at indicated time points (before injection: Day-1 (D−1), 2 and 6 days after injection: Day+2 (D+2) and Day+6 (D+6)]. As shown in Figure 2, on day 6 after hASCs transplantation, we found some dark signals for hASCs labeled with MNP in brains of 6-OHDA induced PD mice transplanted with hASCs compared with other groups. In particular, we observed relatively dark signals in the substantia nigra of 6-OHDA/ASC-MNP group mice 6 days after hASCs transplantation (the area outlined in red on the D+6 image) when compared with before images (D−1 and D+2) (Figure 2 & Appendix A). In other mice, no meaningful change possibly occurred before and after injection.

### 2.3. In Vivo Fluorescence Imaging of hASCs Using Magnetic Nanoparticles in 6-OHDA Induced PD Mouse

To examine the organ distribution of transplanted hASCs, Maestro imaging was performed in Sham/Saline, Sham/ASC-MNP, 6-OHDA/MNP, and 6-OHDA/ASC-MNP groups (Figure 3). We observed that hASCs labeled with magnetic nanoparticles are mostly distributed in brain, lung, and kidney of mice (Figure 3A,B). In particular, brain distribution was prominent in 6-OHDA/ASC-MNP group (Figure 3C). Furthermore, the average brain signal intensity showed the highest increase in the 6-OHDA/ASC-MNP group compared to the other groups (Figure 3D).

### 2.4. Effect of hASCs Using Magnetic Nanoparticles on Behavior Performance of 6-OHDA Induced PD Mice

In the present study, we performed behavioral assessments using the apomorphine-induced rotation (Figure 4A,C) and rotarod tests (Figure 4B,D). In the apomorphine-induced rotation test that evaluates nigrostriatal damage, contralateral net turns per minute were significantly higher in the 6-OHDA-induced PD group (6-OHDA/Saline) than those in the sham groups (Sham/Saline and Sham/ASC-MNP) (Figure 4A,C). hASCs transplantation significantly decreased contralateral net turns per minute in 6-OHDA induced PD mice (6-OHDA/ASC-MNP) on day 33 after hASCs injection (Figure 4C), but not on day 12 (Figure 4A). Moreover, in the rotarod test used to assess motor coordination and balance, hASCs transplantation showed significant recovery in latency time in 6-OHDA/ASC-MNP group on day 33 after hASCs injection (Figure 4D), but not on day 12 (Figure 4B).

## 3. Discussion

In the present study, we investigated whether the transplantation of hASCs using magnetic nanoparticles recovers motor functions through targeted brain distribution in a 6-OHDA induced PD mouse model.

MSCs are adult stem cells with the potential for self-renewal and can differentiate into various tissues such as bone, cartilage, adipose, glial cells, and neurons [4]. Furthermore, MSCs reportedly induce the expression of tyrosine hydroxylase and neurotrophic factors and improve behavioral performance in PD animal models [25,26,27]. ASCs, a type of MSCs, may be potential candidates for stem cell-based PD therapy [13,25]. Reportedly, hASCs increase the secretion of neurotrophic factors such as brain-derived neurotrophic factor (BDNF), nerve growth factor (NGF), and glial-derived neurotrophic factor (GDNF) in stem cell differentiation, neuronal survival, and growth [14,28]. Furthermore, hASCs can differentiate into motor neuron-like cells [11,12,14]. The active motor neuron-like cells that were differentiated from hASCs enhance the expression of specific motor neuron markers such as microtubule-associated protein 2 (MAP2), homeobox 9 (HB9), and choline acetyltransferase (ChAT), and reduce injury cavities in a spinal cord injury mouse model [12].

Magnetic nanoparticles can be used as effective carriers for targeted delivery in the brain [18,19]. This study revealed that hASCs transplantation using magnetic nanoparticles increased distribution to the brain, particularly the substantia nigra. It is well known that the major pathologic hallmark of PD is the loss of dopaminergic neurons in the substantia nigra [1,3]. For magnetic nanoparticles that cross the blood-brain barrier, the concentration and distribution can be regulated in targeted regions using eternal driving magnets [19,29]. Furthermore, magnetic nanoparticles can be imaged and monitored using in vivo imaging technology after loading hASCs [20,21]. A previous study has reported that the distribution of magnetic nanoparticle clusters was significantly associated with damage to the cell membrane [30]. Damaged dopaminergic neurons may accelerate the distribution of magnetic nanoparticle-based hASCs. Interestingly, recent reports showed that magnetic nanoparticles cause stem cells to differentiate into neurons and improve survival and growth of neurons [22,23]. Thus, the distribution of targeted hASCs using magnetic nanoparticles may be crucial in the treatment and management of PD, although further research is needed to elucidate related mechanisms.

In the present study, motor function in PD model mice was found to recover via targeted distribution of hASC. A previous study showed that behavioral performance in a PD mouse model was ameliorated at 3 weeks after hASCs transplantation [13]. Another study has reported that hASCs transplantation restores dopaminergic cell numbers at 22 days in an MPTP-induced PD mouse model, despite improving behavioral performance at an earlier time point [25]. Our MRI results revealed that the transplanted hASCs migrated to the brain and distributed in the substantia nigra 6 days after the hASCs injection. The previous study revealed no detectable GFP-positive cells in the brain at 4 weeks after transplantation [31]. These results prove that transplanted hASCs are delivered to the brain in small amounts and remain there only for a short time. However, impaired motor function in 6-OHDA-induced PD mouse model was significantly improved at 33 days after administration of hASCs, but not at 12 days. After behavioral tests, we confirmed the recovery of nigrostrial dopamine neurons by hASC transplantation in the 6-OHDA/ASC-MNP mice. Dopaminergic neuronal cell death in the SN was sharply increased following the injection of 6-OHDA, and the dopaminergic neurons recovered with the hASC transplantation (Appendix A). Although the therapeutic mechanism of hASC transplantation remains unclear until now, several neurotrophic factors (NTF) secreted from transplanted hASC are believed to affect the neuroprotective and neuro-restorative effects of dopaminergic neurons [32].

Further research is warranted to determine a suitable time point for effective distribution into the substantia nigra, as well as for motor function recovery in PD; however, this study suggested that a specific period is required for motor function recovery after distribution in the substantia nigra.

Therefore, we revealed that hASCs transplantation recovered motor function through targeted brain distribution in a 6-OHDA-induced PD mouse model. These results indicate that hASCs transplantation using magnetic nanoparticles may be a potential therapy for PD.

## 4. Materials and Methods

### 4.1. Preparation of Animals

Seven-week-old C57BL/6N (n = 15 mice per group) male mice were assigned to following 4 groups for behavior tests: Sham mice injected with saline (Sham/Saline), Sham mice injected with magnetic nanoparticles-labeled hASCs (Sham/ASC-MNP), 6-OHDA PD mice injected with saline (6-OHDA/Saline), and 6-OHDA PD mice injected with magnetic nanoparticles-labeled hASCs (6-OHDA/ASC-MNP). Furthermore, we used the following four groups (Sham/ASC-MNP, 6-OHDA/Saline, 6-OHDA/MNP, and 6-OHDA/ASC-MNP groups) for MR images, and the following four groups (Sham/Saline, Sham/ASC-MNP, 6-OHDA/MNP, and 6-OHDA/ASC-MNP groups) for Maestro images. The mice were injected with 10 μg 6-OHDA (4 μg/μL containing 0.2 mg/mL L-ascorbic acid) or equal volumes of saline, unilaterally into the substantia nigra (A/P = −3.2, M/L = −1.5, D/V = −4.6) using a Kopf stereotaxic frame (Kopf Instruments, Tujunga, CA, USA) to establish the PD model or sham control according to the previous studies [13]. All animal procedures were performed according to the National Institutes of Health guidelines for the Humane Treatment of Animals and approved by the Institutional Animal Care and Use Committee of Seoul National University (IACUC No. SNU1001819-1), Seoul National University Hospital (IACUC No. SNUH12-0369) and Gachon University (IACUC No. LCDI-2019-0089).

### 4.2. Preparation of hASCs Using Magnetic Nanoparticles

According to previous studies, hASCs were isolated from human adipose tissues obtained from the lower abdomen under Good Manufacturing Practices (GMP) conditions at the Stem Cell Research Center of RNL BIO (Seoul, Korea) with approval from the Institutional Review Board of the ASAN Medical Center (IRB No.2006-0308) [13,33]. The obtained hASCs were labeled with a magnetic silica core-shell nanoparticle, NEO-LIVE^TM^-Magnoxide 797 (BITERIALS Co., Ltd., Seoul, Korea), which contains Near-IR fluorescent dyes. Particle size is 50 nm (±10%). hASCs were incubated for 24 h in a growth medium containing magnetic nanoparticles at a concentration of 0.2 mg/mL, and then washed with phosphate-buffered saline. After collecting the magnetic nanoparticle-labeled hASCs (1 × 10^6^), they were administered by tail vein injection. All procedures were performed according to the manufacturer’s instructions and previous study [33].

### 4.3. In Vivo Imaging Using Magnetic Resonance and Maestro Imaging System

MR images were obtained using the BioSpec^®^ 94/20 USR AV III system, operated with ParaVision^®^ 6.0 (Bruker BioSpin Corporation, Billerica, MA, USA). The system was equipped with a 210-mm horizontal bore main magnet, and a 400-mT/m actively shielded gradient with integrated shim coils. Furthermore, fluorescence imaging of hASCs using magnetic nanoparticles was performed using Maestro imaging system (CRI Inc., Woburn, MA, USA). The day before imaging, mice hair was removed by shaving from the dorsal and ventral sides. Sequential Maestro images were obtained 9 days after hASCs injection to assess the distribution of transplanted hASCs. Then, all organs were extracted and captured using the Maestro imaging system. In this study, we used hASCs-injected mouse (Sham/ASC-MNP), 6-OHDA induced PD mouse (6-OHDA/Saline), magnetic nanoparticle-injected 6-OHDA PD mouse (6-OHDA/MNP), hASCs-injected 6-OHDA induced PD mouse (6-OHDA/ASC-MNP) groups for MR images, and hASCs-injected mouse (Sham/ASC-MNP), magnetic nanoparticles-labeled 6-OHDA induced PD mouse (6-OHDA/MNP), and hASCs-injected 6-OHDA induced PD mouse (6-OHDA/ASC-MNP) groups for Maestro images.

### 4.4. Behavioral Tests

To evaluate motor functions, behavioral tests, including apomorphine-induced rotation and rotarod, were performed on day −2, +12, and +33 (1, 3, and 6 weeks after 6-OHDA in saline-injected mouse (Sham/Saline), hASCs-injected mouse (Sham/ASC-MNP), 6-OHDA induced PD mouse (6-OHDA/Saline), and hASCs-injected 6-OHDA induced PD mouse (6-OHDA/ASC-MNP) group. The apomorphine-induced rotation test, to assess the hypersensitivity of the lesioned striatum, was performed after subcutaneous apomorphine administration (0.05 mg/kg). The mice were tracked, and the number of right and left rotations and the net number of rotations during 30 min were measured using the EthoVision video tracking system (Noldus, Wageningen, The Netherlands). The accelerating rotarod test was performed to evaluate motor coordination and balance. All mice were conditioned at a speed of 8 rpm for 5 min and a speed of 12 rpm for 5 min, at an interval of 1 h in the rotarod apparatus (B.S. Technolab Inc., Seoul, Korea). The day after the training, the motor performance was assessed at an accelerated speed from 2 to 20 rpm for 10 min. The rotarod test of each mouse was calculated by averaging the latency time measured in 3 trials.

### 4.5. Statistical Analysis

All statistical analyses were performed using GraphPad Prism version 8.4.1 (GraphPad Software Inc., San Diego, CA, USA). All data are presented as the mean ± standard error of the mean (SEM). Differences between groups in all collected data were analyzed using one-way analysis of variance followed by the Bonferroni post hoc test, and Kruskal–Wallis was employed for non-parametric evaluation. A *p*-value of <0.05 was considered statistically significant.

## Figures and Tables

**Figure 1 ijms-22-00654-f001:**
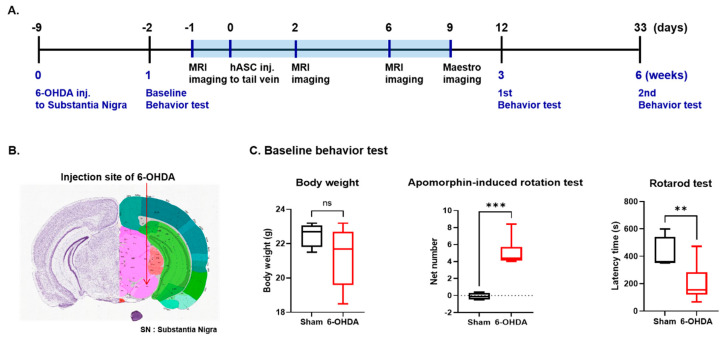
Experimental procedures. (**A**) Experimental scheme for in vivo imaging and behavioral tests. (**B**) The 6-OHDA was stereotaxically injected into the Substance Nigra (SN) (AP, −3.2 mm; ML, −1.5 mm; DV, −4.6 mm). The coronal section from the Allen Brain Atlas shows the SN region (arrow). (**C**) Baseline behavioral tests, including body weight, apomorphine-induced rotation test, and rotarod test, were performed in saline-injected control mouse (Sham) and 6-OHDA-induced PD mouse (6-OHDA) 1 week after 6-OHDA injection (n = 10~12 per group). ** *p* < 0.01, *** *p* < 0.001. SN, Substantia nigra; 6-OHDA, 6-hydroxydopamine; PD, Parkinson’s disease.

**Figure 2 ijms-22-00654-f002:**
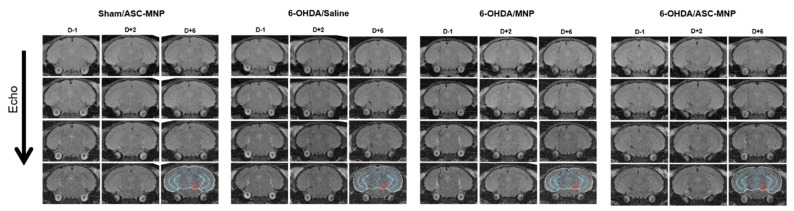
Sequential MR imaging was performed in mice injected with NEO-LIVETM-Magnoxide 797-labeled hASCs. MR images were acquired at optimized parameters on a 9.4 T Bruker animal imager at the indicated time points. MR imaging parameters were follows; Pulse sequence = Multiple Gradient Echo 2D, Echo train = 10, Selected image’s Echo train = 2, Echo spacing = 3 ms, Effective TE (Echo time) = 6 ms, TR (Repetition time) = 2000 ms, FA (Flip Angle) = 50 degree, Matrix size = 256 × 256, Slice thickness = 0.7 mm, FOV (Field of view) = 15.36 × 17.92 mm^2^, Pixel bandwidth = 520.83 pixel/Hz. Sham/ASC-MNP; Sham mice injected with magnetic nanoparticles-labeled hASCs, 6-OHDA/Saline; 6-OHDA PD mice injected with saline, 6-OHDA/MNP: 6-OHDA PD mice injected with magnetic nanoparticles, 6-OHDA/ASC-MNP: 6-OHDA PD mice injected with magnetic nanoparticles-labeled hASCs. MR, magnetic resonance; hASCs, human adipose-derived stem cells; 6-OHDA, 6-hydroxydopamine; MNP, magnetic nanoparticles; PD, Parkinson’s disease.

**Figure 3 ijms-22-00654-f003:**
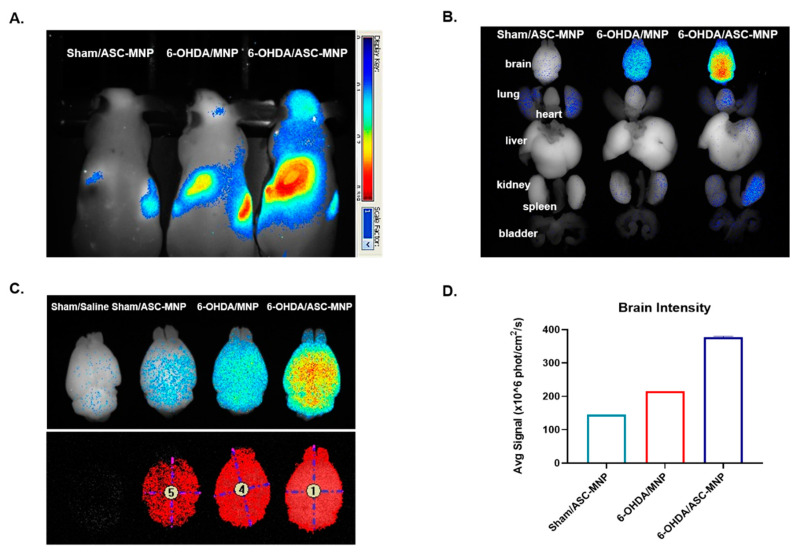
Maestro imaging was performed in hASCs injected mouse (Saline/ASC-MNP), magnetic nanoparticles-labeled 6-OHDA induced PD mouse (6-OHDA/MNP), hASCs-injected 6-OHDA induced PD (6-OHDA/ASC-MNP) mouse. Fluorescent magnetic NEO-LIVETM-Magnoxide 797 nanoparticles were administered at 0.4 mg/mL by tail vein injection for in vivo imaging. (**A**) Maestro imaging in living Sham/ASC-MNP, 6-OHDA/MNP, and 6-OHDA/ASC-MNP mouse. (**B**) Organ distribution of magnetic nanoparticles in Sham/ASC-MNP, 6-OHDA/MNP, and 6-OHDA/ASC-MNP mouse. (**C**) Brain distribution of magnetic nanoparticles in Sham/Saline, Sham/ASC-MNP, 6-OHDA/MNP, and 6-OHDA/ASC-MNP mouse. (**D**) Brain intensity of magnetic nanoparticles in Sham/ASC-MNP, 6-OHDA/MNP, and 6-OHDA/ASC-MNP mouse. hASCs, human adipose-derived stem cells; 6-OHDA, 6-hydroxydopamine; MNP, magnetic nanoparticles; PD, Parkinson’s disease.

**Figure 4 ijms-22-00654-f004:**
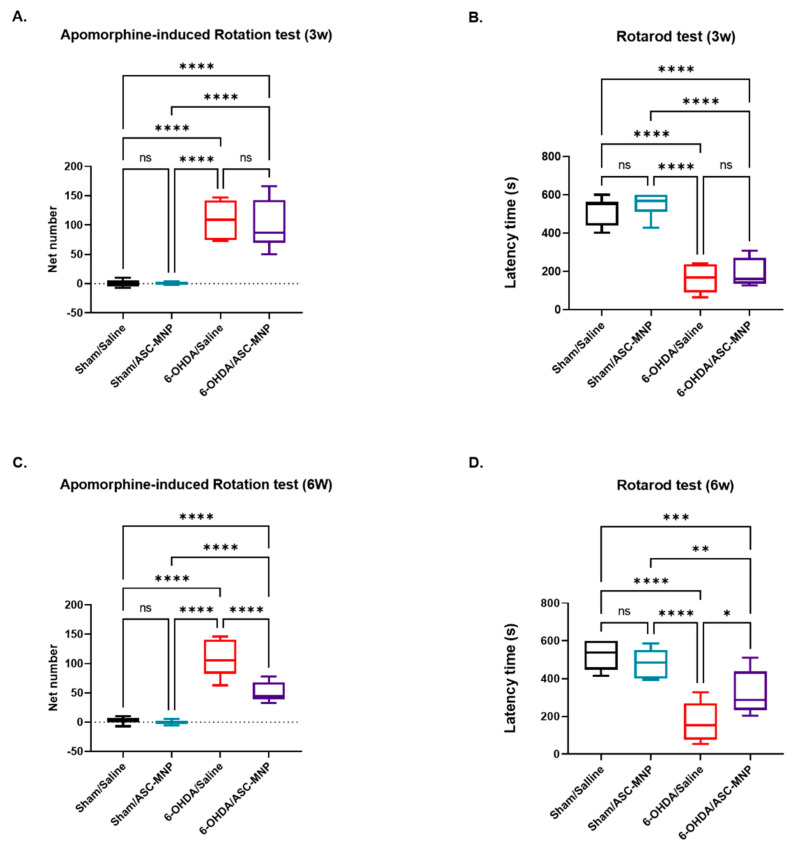
Behavioral tests at 3 and 6 weeks after 6-OHDA injection. (**A**) Apomorphine-induced rotation test was performed 3 weeks after 6-OHDA administration (12 days after hASCs injection) in all groups. (**B**) Rotarod test was conducted 3 weeks after 6-OHDA administration (12 days after hASCs injection) in all groups. (**C**) Apomorphine-induced rotation test was performed 6 weeks after 6-OHDA administration (33 days after hASCs injection) in all groups. (**D**) Rotarod test was conducted 6 weeks after 6-OHDA administration (33 days after hASCs injection) in all groups. All data represent the mean ± standard error of the mean (n = 8 per group). * *p* < 0.05, ***p* < 0.01, ****p* < 0.001, **** *p* < 0.0001. 6-OHDA, 6-hydroxydopamine; hASCs, human adipose-derived stem cells; MNP, magnetic nanoparticles; PD, Parkinson’s disease. Experimental groups: Sham/Saline; saline-injected sham mouse, Sham/ASC-MNP; hASCs-injected sham mouse, 6-OHDA/Saline; saline-injected 6-ODHA induced PD mouse, 6-OHDA/ASC-MNP; hASCs-injected 6-ODHA induced PD mouse.

## Data Availability

Not applicable.

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
