# Peer review of "Therapeutic Potential of Magnetic Nanoparticle-Based Human Adipose-Derived Stem Cells in a Mouse Model of Parkinson’s Disease"

_ijms, 2021, doi:10.3390/ijms22020654_

Round 1

Reviewer 1 Report

In this manuscript the authors show that labelling stem cells with magnetic nanoparticles has beneficial effects on their biodistribution, especially in the brain. This imaging study (MRI, fluorescence) is completed by behavioral tests that indicate that mice treated with magnetically labeled cells show fewer symptoms of Parkinson's disease in comparison with non treated cells.

However, the authors do not provide convincing explanations for the results obtained. Why is it usefull to label the stem cells with magnetic nanoparticles ? Have they use magnetic targeting, for example, using a permanent magnet placed on the mouse head? Have they performed a control with nonmagnetically labeled stem cells in behavorial tests ? How do they explain a difference in magnetically labeled and  nonmagnetically labeled stem cells if they have not use magnetic targeting ? They have to clarify all those points.

The presentation of the results is also a bit confusing, for example in section 2.2. 

It is indicated that the MRI studies were done on 4 groups of animals while results are presented for 3 types of animals (hASCs-injected wild type, 6-OHDA induced PD, and hASCs-injected 6-OHDA induced PD 85 mice), using different "protocols": hASCs-injection for wild type and 6-OHDA induced PD mices, and saline or "nano" for  6-OHDA induced PD mices. 

The different protocols should be more clearly explained, e.g. what does the "nano" protocol mean: is it an injection of nanoparticles that have not been incubated with cells?

I also wonder if it would not be clearer to compare only the transplantation of hASCs cells between wild type and 6-OHDA induced PD mices. Controls (saline , "nano" for for  6-OHDA induced PD mices) could be afforded in supplementary materials.

Similarly, the presentation of results in section 2.3 and 2.4 could be improved by focusing on the most significant results and comparisons. In section 2.4, figure 4, the authors have to clarify what is "sham" mices.

Finally, it seems to me essential that the authors give more precision concerning the nanoparticles used (physical size, hydrodynamics, magnetic properties, surface functionalization....) and the conditions of incubation with the cells (medium, particle concentration, quantity of cells ...). Do the authors have experimental evidence that nanoparticles have been internalized by stem cells (confocal microscopy ? TEM image of cell sections obtained by microtomy ?) ?

Reviewer 2 Report

The manuscript entitled “Therapeutic potential of magnetic nanoparticle-based human adipose-derived stem cells in a mouse model of Parkinson’s disease” by Kim KY and Chang K-A demonstrates that human adipose-derived stem cells (hASC) accumulate in the brain of mice treated with 6-hydroxydopamine which induces Parkinson’s disease. The loading of the hASCs with fluorescently labelled magnetic nanoparticles allows the detection of the hASCs by MRI and with the Maestro system. The authors could also clearly show that the treated mice exhibit an improved behavioural performance after 6 weeks. For these investigations, the authors used the apomorphine-induced rotation test as well as the rotarod test.

The manuscript contains new and interesting data. The experimental setting is accurately planned and the experiments are well conducted and presented. Nevertheless, the goal of the manuscript seems to be over interpreted.

I suggest accepting the paper after major revision.

Concerns:

I am not sure, why the authors label the hASCs with magnetic nanoparticles. The aim of the manuscript should be “to investigate the therapeutic potential of hASCs using magnetic nanoparticles”. As far as I understand the paper, the authors used the labelling only for detection purposes. The magnetic nanoparticles do not contribute to the therapeutic effect of the hASCs. There is no benefit whether the hASCs are loaded with MNPs or not. That would be for example the case when the magnetic nanoparticles are also used for directing the hASCs to the brain. The authors should clarify that point.

Minor comments:

The quality of figure 2 is poor. I could not see any difference in the MR images of D+6. The authors should improve that figure or add an addition figure highlighting the images from D+6.

The hASCs were applied in the tail vein. How is the distribution in the organs of the mice? I would expect high levels in the liver.

Round 2

Reviewer 1 Report

I would like to thank the authors for greatly improving the presentation of their results.

I suggest adding the words "MNP labeled" to line 67 to improve this presentation (MNP labeled - hASCs-injected 67 sham (Sham/ASC-MNP)).

Thus it will be homogeneous with what is indicated line 68-69 (MNP labeled  hASCs-injected 6-OHDA-induced PD (6-OHDA/ASC-MNP)).

And then it will be now easy to understand in the whole manuscript what are Sham/ASC-MNP and 6-OHDA/ASC-MNP mices.

Some mistakes can also be corrected : 

-line 72 : MNP instead of NMP

-line 162 : MNP instaed of MN

-line 220 : MNP instead of NMP

-line 250 : MNP instead of NMP

Author Response

Thank you for your comments. According to your recommendation, we corrected our mistakes in the revised manuscript in red. 

Point 1: I suggest adding the words "MNP labeled" to line 67 to improve this presentation (MNP labeled - hASCs-injected 67 sham (Sham/ASC-MNP)).

Thus it will be homogeneous with what is indicated line 68-69 (MNP labeled  hASCs-injected 6-OHDA-induced PD (6-OHDA/ASC-MNP)).

And then it will be now easy to understand in the whole manuscript what are Sham/ASC-MNP and 6-OHDA/ASC-MNP mices.

Response 1: Thank you for your comments. According to your recommendation, we added the words "MNP labeled" in the revised manuscript in red. 

Point 2: Some mistakes can also be corrected : 

-line 72 : MNP instead of NMP

-line 162 : MNP instaed of MN

-line 220 : MNP instead of NMP

-line 250 : MNP instead of NMP

Response 2: Thank you for your comments. According to your recommendation, we corrected our mistakes in the revised manuscript in red. 

Reviewer 2 Report

The authors have carefully addressed the reviewer's comments and the manuscript can be published as is.

Author Response

On behalf of my co-authors, we thank you very much for giving us positive and constructive comments and suggestions on our manuscript.